# The Relative Positioning of B and T Cell Epitopes Drives Immunodominance

**DOI:** 10.3390/vaccines10081227

**Published:** 2022-07-31

**Authors:** Riccardo Biavasco, Marco De Giovanni

**Affiliations:** 1Department of Genetics and Cell Biology, IRCCS San Raffaele Scientific Institute, 20132 Milan, Italy; riccardo.biavasco@gmail.com; 2Department of Microbiology and Immunology, University of California San Francisco, San Francisco, CA 94143, USA

**Keywords:** antibody, immunodominance, viral escape, epitope

## Abstract

Humoral immunity is crucial for protection against invading pathogens. Broadly neutralizing antibodies (bnAbs) provide sterilizing immunity by targeting conserved regions of viral variants and represent the goal of most vaccination approaches. While antibodies can be selected to bind virtually any region of a given antigen, the consistent induction of bnAbs in the context of influenza and HIV has represented a major roadblock. Many possible explanations have been considered; however, none of the arguments proposed to date seem to fully recapitulate the observed counter-selection for broadly protective antibodies. Antibodies can influence antigen presentation by enhancing the processing of CD4 epitopes adjacent to the binding region while suppressing the overlapping ones. We analyze the relative positioning of dominant B and T cell epitopes in published antigens that elicit strong and poor humoral responses. In strong immunogenic antigens, regions bound by immunodominant antibodies are frequently adjacent to CD4 epitopes, potentially boosting their presentation. Conversely, poorly immunogenic regions targeted by bnAbs in HIV and influenza overlap with clusters of dominant CD4 epitopes, potentially conferring an intrinsic disadvantage for bnAb-bearing B cells in germinal centers. Here, we propose the theory of immunodominance relativity, according to which the relative positioning of immunodominant B and CD4 epitopes within a given antigen drives immunodominance. Thus, we suggest that the relative positioning of B-T epitopes may be one additional mechanism that cooperates with other previously described processes to influence immunodominance. If demonstrated, this theory can improve the current understanding of immunodominance, provide a novel explanation for HIV and influenza escape from humoral responses, and pave the way for a new rational design of universal vaccines.

## 1. Introduction

Antibodies are a fundamental component of human immunological defense, and one of their most important functions is to confer protection against viruses and exogenous microorganisms upon primary infection or vaccination [1,2,3]. Antibodies that are able to provide sterilizing immunity, by preventing pathogen entry and interaction with target cells, are commonly called ‘neutralizing antibodies’ (nAbs) [1]. nAbs are produced by B cells that have been selected in the host germinal centers (GCs) [3,4,5]. Upon antigen encounter, antigen-specific B cells start proliferating, interact with cognate CD4 T cells, and then migrate to the center of B follicles where they establish the GC microstructure [3,4,6,7,8]. GCs are the microanatomical niches where B cell clones mature, receive survival signals by cognate CD4 T cells, mutate their B cell receptors (BCRs), and are selected to produce high-affinity nAbs [3,5,9].

The repertoire of naïve B cells and BCRs is extremely diverse thanks to BCR rearrangement during B cell development, resulting in the potential to produce antibodies against virtually any epitope of a given antigen [2,10,11]. Despite this potential, epitope specificities are not equally targeted by humoral responses, with the most frequently targeted epitopes defined as immunodominant [3,12]. Following immunization, the higher prevalence of specific B and T cell clones, which expand at the expense of other epitope-specific cells, is referred to as immunodominance [3,12]. The combination of germline B cell precursor frequency, antigen accessibility, affinity/avidity, and CD4 T cells help influence the Darwinian selection process of B cell clones in GCs and largely impact epitope immunodominance [1,3].

The goal of most vaccines is to induce the vigorous and long-term production of nAbs with the ability to prevent any future infection. However, some viruses deploy diverse strategies to escape antibody neutralization, including the mutation of viral antigens (e.g., HIV and influenza) [1,3,13]. Antibodies that bind to conserved viral epitopes and are able to neutralize different viral mutants and strains are referred to as ‘broadly neutralizing antibodies’ (bnAbs) [1]. Conserved viral epitopes often represent the ‘Achilles heel’ of mutating viruses, as they cannot be mutated without altering important steps in the viral life cycle [1,14,15]. While the induction of bnAbs represents the major goal of most vaccination approaches, none of the strategies tested to date have successfully and consistently induced anti-HIV and anti-influenza bnAbs [13,15,16]. Indeed, the immunodominance of B cell clones specific for variable and/or non-broadly neutralizing viral epitopes has recently proven to be a major obstacle in vaccine design, with the enhanced production of non-neutralizing antibodies at the expense of the bnAbs [3,12,15,16].

Poor or defective bnAbs induction following vaccination in the context of HIV and influenza has been extensively studied. The following potential explanations for defective bnAbs production (e.g., anti-gp120 CD4 binding site for HIV, anti-stem of the hemagglutinin (HA) for influenza) have been proposed: poor epitope accessibility [3,17,18], high mutational load required to generate bnAbs [1], low neutralizing B cell precursor frequency [19], and HLA-II polymorphisms [20,21]. Nevertheless, none of these arguments can fully recapitulate the apparent counter-selection for bnAbs, with recent experimental evidence arguing against such theories.

Around 30 years ago, antibody binding was shown to influence antigen processing and presentation, with CD4 T-cell epitopes either inhibited or boosted based on their relative positioning to the antibody epitope [22,23,24]. Here, we propose the theory of immunodominance relativity, according to which the relative positioning of B and T cell epitopes within a given antigen drives immunodominance. During a global pandemic caused by a newly emerged coronavirus, understanding the molecular bases of immunodominance has become of paramount importance, particularly to guide the rational design of future universal vaccines.

## 2. Historical Viral Escape Theories

In recent decades, several hypotheses have been proposed to explain the molecular mechanisms underlying the inconsistent induction of bnAbs, especially in the context of HIV and influenza vaccination studies.

### 2.1. Antigen Variability and Epitope Accessibility

The variation of immunodominant epitopes due to the accumulation of random mutations in the viral genome represents one of the most successful strategies for viruses to escape the host immune system [1,13]. This strategy is particularly relevant for RNA viruses, which rely on more error-prone molecular machinery when duplicating their genome [25]. However, HIV and influenza envelope proteins (gp120 and HA, respectively) contain three-dimensional structures that are not permissive to a high mutational load, as this would result in the impairment of essential steps during the viral life cycle, such as binding to target receptors, membrane fusion, and internalization [1,25,26]. Indeed, conserved regions of these antigens, namely, the CD4 binding domain of gp120 and the stem of HA, contain the epitopes targeted by the most potent bnAbs described to date against these two viruses, 3BNC117 [27] and MEDI8852 [28], respectively. Nevertheless, antibody responses targeting the CD4 binding domain and the stem of HA are very rare in the general population.

Epitope accessibility has long been regarded as a fundamental requirement to effectively mount humoral responses [1,14,18,29,30,31,32]. First-generation anti-HIV bnAbs targeting the CD4 binding site showed limited neutralization breadth and/or potency, a finding thought to be related to the poor accessibility of these epitopes in HIV-1 primary isolates [33]. Recently, however, a renewed experimental effort has led to the isolation of several new anti-CD4 binding site human monoclonal antibodies (VCR01-like) characterized by greater breadth and potency [19,34,35]. Moreover, the CD4 binding site is accessible for binding to human CD4 molecules [14], an essential step during viral entry into target cells, and VRC01-like Abs can be generated in macaques following vaccination [36], indicating that this region can be immunogenic in vivo. In the context of influenza HA, steric hindering from the head domain has been considered to prevent Ab binding to the stem domain [1,15]. Nevertheless, further increasing the accessibility of the stem with head-less HA immunogens did not result in potent bnAbs induction [1,37], suggesting overall that the low accessibility of conserved viral epitopes does not fully explain mechanisms for subdominant bnAbs induction. More recently, bnAbs were shown to efficiently bind to cryptic epitopes in different microbial antigens, such as coronavirus [38], ebolavirus [39], and plasmodium [40], conferring neutralization across different subspecies of those pathogen families.

### 2.2. B Cell Precursor Frequency, Somatic Hypermutation, and HLA2 Polymorphisms

The frequency of germline B cell precursors shapes the humoral response to immunogenic antigens by impacting GC occupancy [1,3]. A low B cell precursor frequency has been proposed as one potential obstacle to mounting effective bnAb responses following vaccination. Despite being relatively rare in the repertoire, VRC01-like cells can be activated by a high-affinity stimulation in mouse models that recapitulate human precursor frequencies [3,18]. VRC01-like precursors are also present in 96% of humans [19], giving hope that a bnAb-like HIV vaccine is indeed possible. Furthermore, germline residues of the VH1-69 alleles, which account for 2–6% of the B cell repertoire, are known to mainly mediate recognition of influenza group 1 HA stem [1,41]. This extraordinary high number of putative anti-stem B cell precursors correlates, however, with a subdominant VH1-69 response that is generated only in some individuals.

The high mutational load required to develop bnAbs can also represent a major challenge [1]. Indeed, anti-HIV VRC01-like antibodies require multiple rounds of somatic hypermutation to be generated and are frequently characterized by extensive nucleotide insertions and substitutions [1,3]. Nonetheless, it was recently shown that a single proline-to-alanine mutation in HCDR2 was sufficient to confer high-affinity binding to the influenza HA stem [41], consistent with a rapid affinity maturation process. Despite the low number of mutations required, anti-influenza bnAbs are rare and apparently counter-selected, suggesting that high mutational load and complex antibody selection processes do not fully recapitulate defective bnAbs induction, at least in the context of influenza.

A variety of host genetic factors can influence the outcome of viral infections, most notably polymorphisms within the HLA class I and II loci [20,42,43,44]. HLA class II molecules are essential in the development of adaptive immune responses, as they present antigens to CD4 T cells and contribute to the regulation of B and T cell interactions within GCs. HLA DRB1*13-DQB1*06 was associated with a trend toward the increased duration of AIDS-free time in HIV patients treated with anti-retroviral therapy [43]. In addition, the inheritance of DRB1*13 alleles has been associated with long-term survival among children with vertically transmitted HIV-1 infection [20] and higher IFNγ production by Th1 cells [21]. Nevertheless, anti-CD4 binding site (and gp41-MPER) bnAb responses were detected in very few individuals independently of HLA class II haplotypes [45], suggesting that HLA class II polymorphisms are not likely to explain the defective anti-viral bnAbs production following vaccination.

### 2.3. BCR and Antibody Modulation of Antigen Presentation

Antigen processing and presentation were initially investigated using macrophages or dendritic cells as a model of antigen presenting cells (APCs) [46]. In such cases, the APCs do not possess specific antigen receptors and antigen internalization is mainly restricted to receptor-free endocytosis. In contrast, antigen uptake by antigen-specific B lymphocytes is mediated by the BCR, which mediates efficient antigen presentation at lower concentrations than those required by non-specific B cells [23,47]. Antibodies can alter the conformation and stability of target antigens [48] and protect epitopes from proteolytic processing [49]. In particular, antibody–antigen complexes can resist the lysosomal acidic pH [22], therefore influencing the antigen fragmentation by proteases, which can initiate while the antigen is still bound by antibodies [50].

Pioneer work by Berzofsky and Celada indicated that antibody binding could differentially boost antigen presentation to some T cell clones, either through the modulation of antigen uptake or proteolytic processing [51,52,53,54]. Accordingly, it was hypothesized that the antibody specificity shapes the initial pattern of antigen fragmentation, supporting the existence of T and B cell preferential pairing [51,55] and reciprocity circuits [51]. A few years later, the sophisticated work from P.D. Simitsek et al. demonstrated that antibody binding to antigens can modulate their processing by enhancing or suppressing HLA class II presentation of different CD4 T cell determinants [56]. Strikingly, a single bound antibody (11.3) or its Fab fragment was shown to simultaneously boost the presentation of one CD4 T-cell epitope (1273–1284 aa), while suppressing the in vitro presentation of other determinants (1174–1189 aa, Appendix A) [56]. Both tetanus epitopes that are modulated by BCR/antibody binding were shown to fall within the 11.3 Ab footprint region (i.e., the region of the antigen that is at least partially protected from lysosomal proteolytic cleavage). The suppressed epitope (1174–1189 aa) is sterically hindered to bind to HLA class II molecules upon interaction with 11.3 Ab, likely due to reduced proteolysis in the lysosomal and inaccessibility to HLA binding [56]. On the contrary, the boosted epitope (1273–1284aa) was likely protected from excessive cleavage, stabilized, and made more accessible for the binding to HLA class II molecules by the interaction with the 11.3 Ab. The processing of epitopes located far from the BCR/antibody binding region (947–967 aa) was instead maintained unaltered. The influence of the antibody specificity on antigen degradation was further analyzed in clones of tetanus-specific B cells with different epitope specificities [50,57], and was confirmed using other model antigens, such as f3-galactosidase [52] and myoglobin [54].

## 3. Results

### 3.1. Strong Inducers of Neutralizing Humoral Responses

Given the impact of BCR and antibody binding on antigen processing, we postulated that the relative positioning of B and CD4 T cell epitopes shapes immunodominance. To test this hypothesis in the absence of studies that map B and CD4 T cell immunodominant epitopes within single patients, we analyzed the 3D positioning of published B and T cell epitopes in immunogenic antigens known to elicit strong and neutralizing humoral responses, such as measles HA, diphtheria toxoid, vesicular stomatitis virus glycoprotein (VSV-GP), and SARS-CoV-2 spike protein. To restrict our focus to highly immunodominant and HLA-independent regions, we gathered published data for B and T cell epitopes and selected determinants found in multiple patients or host species (mice, macaques, and humans). This approach allowed us to select determinants that were likely to be dominant in the presence of different HLA class II alleles. In addition, we decided to focus our analysis on experimentally validated rather than predicted epitopes [58]. Across measles HA and diphtheria toxoid, immunodominant CD4 epitopes are scattered throughout the protein sequences and often adjacent to immunogenic and neutralizing B-cell epitopes (e.g., measles HA: T epitopes 321–350 [59], 443-469 [60]; B epitopes 309–319, 380–400aa [61]; diphtheria toxoid: T epitopes 271–290, 321–340, 411–450 [62], B epitopes 247–260, 395–403, 477–483, 508–527 [63], Figure 1; Appendix A). In the diphtheria toxoid (PDB 1MDT), 5 out of 18 immunodominant B regions (255–260 [64], 381–394 [64], 395–403 [63], 452–458 [64], and 465–475aa [64]) are located next to immunogenic CD4 epitopes (271–290, 351–371, 411–430, 431–450aa [62]), possibly in the ideal position to be boosted upon BCR binding (Figure 1. Interestingly, only 2 out of 18 immunodominant B-cell epitopes (351–355, 409–420aa) [63,64] partially overlap with immunodominant CD4 determinants (351–370, 411–430aa) [62]; however, they are also adjacent to other CD4 determinants (331–350, 421–440aa) [62] (Figure 1 and Appendix A). In VSV-GP (PDB 4YDI), a neutralizing B-cell epitope (382–400aa) [65] is located next to two immunodominant CD4 epitopes (338–368aa) [66] and does not overlap with any CD4 epitope (Appendix A). Protective antibodies targeting the spike protein of SARS-CoV-2 are induced in the majority of infected or vaccinated patients [67,68,69,70,71]. In agreement with the previous observations on immunogenic antigens, four out of eight B-cell immunodominant epitopes (209–226, 721–733, 769–786, 809–826aa) [72,73] are adjacent to immunodominant CD4 determinants (166–180, 751–765, 866–880aa) [74] in the SARS-CoV-2 spike protein, possibly inducing an antigen presentation boost upon BCR binding (Figure 2 and Appendix A). In parallel with other highly immunogenic antigens, none of the immunodominant B regions were found to overlap with dominant CD4 epitopes. Interestingly, 95% of SARS-CoV-2 CD4 epitopes induced upon vaccination were conserved across viral variants [75], suggesting that these concepts may apply to newly emerging viral variants. Despite broad immunogenicity, the spillovers of β-coronaviruses in humans and the emergence of SARS-CoV-2 variants highlight the need for broader anti-coronavirus humoral protection. Recently, a highly conserved B epitope in the stem-helix of SARS-CoV-2 spike (1148–1156aa [69]) was shown to be a target of several bnAbs [68], thus representing a potential target for broad humoral protection. Higher frequencies of anti-stem helix-specific Abs were observed in vaccinated individuals who were previously infected [68], indicating this region is immunogenic in humans. However, these antibodies are found at much lower frequencies in individuals previously infected with SARS-CoV-2 or those who received two doses of mRNA vaccines, indicating that humoral responses targeting the stem helix are usually rare. Interestingly, this highly conserved stem-helix B epitope (1148–1156aa [69]) was not located near any immunodominant CD4 determinant, where antigen presentation boost upon BCR binding is unlikely to occur. This observation can explain the counter-selection of anti-stem-helix bnAbs in favor of more immunogenic but less cross-reactive anti-spike B-cell specificities, which are adjacent to immunodominant CD4 determinants and might take advantage of enhanced antigen presentation. Of interest, other neutralizing and dominant B epitopes were described to locate within the RBD spike domain, distant from dominant CD4 determinants [76,77].

Overall, these data suggest that immunodominant and neutralizing B-cell epitopes are mostly not overlapping with and often adjacent to dominant CD4 T-cell epitopes, increasing the chances of an antigen presentation boost rather than suppression upon BCR binding. Additionally, antigens that are associated with a rapid nAb response contain immunogenic CD4 determinants scattered in several portions of the viral proteins. Scattered CD4 T cell epitopes increase the likelihood for dominant B cell clones targeting multiple regions of the antigen to emerge, facilitating a neutralizing humoral response. The presence of conserved and dominant viral CD4 epitopes that support nAb production raises the intriguing question about why such viruses have evolved to maintain these immune determinants. Some of these epitopes may be part of protein regions that do not allow for a high mutational load, as it would result in a loss of viral fitness. However, this possibility would not explain why such conserved regions contain dominant CD4 epitopes. A second possibility is that these viruses maintained these conserved and dominant CD4 epitopes to obtain long-term fitness advantages. Indeed, this may be particularly the case of aggressive and fast replicating cytopathic viruses (including VSV and measles), which are highly infectious and induce widespread cell damage. The early and potent neutralization of such viruses is key for reaching a balance in the host–pathogen interaction, as hosts incapable of neutralizing such viruses are likely to succumb to the infection. In addition, these viruses usually escape Ab responses by infecting a new host prior to the generation of antiviral adaptive immunity and are thus not highly affected by neutralization. For these reasons, these cytopathic viruses were likely not selected to evolve a protective mechanism against the pressure of the adaptive immune system.

### 3.2. Poor Inducers of Neutralizing Humoral Responses

In contrast, we postulated that the clustering of dominant CD4 T cell epitopes could reduce the immunogenicity of specific antigen portions, suppressing T cell help to B cells targeting those regions. To test this hypothesis, we modeled the relative positioning of B and T cell epitopes in well-studied antigens that efficiently escape broad Ab neutralization, such as HIV gp120 and influenza HA. To minimize the effects of viral antigen variability and HLA class II polymorphisms, we focused on experimentally validated and highly immunodominant CD4 determinants presented by multiple HLAs (see Materials and Methods Section) and conserved B cell epitopes targeted by bnAbs.

The CD4 binding site in HIV gp120 represents, among others, one of the most promising targets to achieve broad anti-HIV protection. Indeed, second-generation anti-CD4 binding-site antibodies (e.g., 3BNC117) broadly neutralize HIV-1 primary isolates and suppress infection upon intravenous injection in chronically infected patients, representing a potent clinical tool. Intriguingly, immunodominant CD4 epitopes are not scattered throughout the whole gp120 protein; rather, they are clustered in the outer domain [78] (Figure 3). In addition, combining rules for HLA class II binding of predicted epitopes to well-conserved sequences substantially improves the prediction of immunodominant CD4 epitopes [79], as epitopes included in conserved sequences are more likely to become immunodominant thanks to a higher frequency across different viral strains. Altogether, these observations suggest that HIV might have evolved to host a cluster of immunodominant CD4 T cell epitopes within the CD4 binding site, a highly conserved region of the gp120 outer domain and the target of the most promising bnAbs (Figure 3). To confirm this observation, we selected validated immunodominant CD4 epitopes published by different research groups, focusing on those shared by multiple HLA class II haplotypes (immunodominant in at least two different mouse strains, macaques, and/or patients). Highly immunodominant CD4 epitopes are localized within three main regions in the gp120 sequence, 300–368 [78,79,80], 400–449 [78,81], and 480–508aa [78,79,81,82,83] (Figure 3 and Appendix A), confirming the clustering of immunodominant epitopes within the CD4 binding site in the outer domain.

Crystal structures of HIV gp120 and 3BNC117 bnAb (PDB 4jpv) are presented in Figure 3. The neutralizing epitope recognized by 3BNC117 is located within the CD4 binding site and overlaps with immunodominant CD4 T cell determinants (300–368, 400–449, and 480–508aa). Strikingly, 62% of the amino acids essential for 3BNC117 binding (Figure 3 and Appendix A) are located within the highly immunodominant CD4 T cell epitopes. A high overlap of B and T cell epitopes may result in the suppression of antigen presentation and intrinsic disadvantage of B cells displaying 3BNC117-like BCRs [19]. In support of this hypothesis, high-affinity anti-CD4 binding site Abs added during APC-Ag pulsing [84] can inhibit gp120-specific CD4 T cell proliferation by suppressing gp120 processing and preventing HLA class II antigen presentation [85]. Moreover, recent experimental evidence suggests that potent help is required to stimulate and expand rare precursors of anti-gp120 bnAbs in vivo, highlighting the importance of T cell help during this process [86].

Similarly, immunodominant CD4 T cell epitopes (401–430aa) [17] largely overlap with neutralizing B cell regions in the influenza HA stem (PDB 5JVR; Figure 4 and Appendix A), with 50% essential amino acids for MEDI8852 binding to the HA stem located within the highly immunogenic CD4 T regions (401–430aa) [17]. In support of the functional role of this relative positioning, anti-stem (but not anti-head) antibodies have recently been shown to specifically inhibit presentation of immunodominant T cell epitopes located within the HA stem [17]. The influenza HA head is the target of most strain-specific nAbs that commonly lack broadly neutralizing activity. Anti-HA-head B cell clones largely dominate the humoral responses, allowing for strain-specific nAbs to emerge in virtually every infected host. Of note, the influenza HA head also contains immunodominant CD4 epitopes (centered around 215 and 265 aa [87]), which are mostly adjacent to immunodominant B cell regions (Figure 4 and Appendix A). This relative positioning recapitulates the structure observed in highly immunogenic antigens, likely leading to enhanced antigen presentation upon BCR binding. On the contrary, the surface of the HA-stem region is constituted of immunodominant CD4 T cell epitopes, maximizing the likelihood of suppressing any B cell clone targeting this conserved protein region, particularly in the context of concomitant presence in the GC reaction.

To quantify the observed differences in the relative positioning of B and T cell epitopes, we grouped together antigens that were strong or poor inducers of long-lasting humoral responses and measured the frequency of B epitopes that were distant from (>15aa), adjacent to (<15aa), or overlapping (0aa) with CD4 T cell epitopes. In support of our hypothesis, the relative positioning of B and T cell epitopes is highly associated with the immunogenicity of the antigens analyzed (*Χ*^2^ = 23.0485, *n* = 66, *p* < 0.0001, Figure 5a and Appendix A), with poor immunogens showing an increased frequency of B epitopes overlapping with immunodominant CD4 determinants compared to strong nAb inducers (58.6% vs. 8.1%, respectively).

Altogether, this analysis supports the immunodominance relativity model, according to which the relative positioning of B and T cell epitopes within antigens drives B cell immunodominance. Epitopes targeted by bnAbs often overlap with highly immunodominant CD4 T cell epitopes within viral antigens that escape Ab neutralization (Figure 5b). Epitope overlap may result in the suppression of antigen presentation, limiting T cell help and introducing an intrinsic disadvantage for B cell clones displaying bnAb-like BCRs. To maximize this effect, immunodominant CD4 T cell epitopes are mostly scattered across the viral antigens that induce efficient Ab neutralization, whereas they tend to cluster within poorly immunogenic regions in antigens that escape humoral responses. In conclusion, the immunodominance relativity model offers an innovative explanation for HIV and influenza escape from long-lasting immunity and the molecular basis of antibody selection and maturation.

## 4. Discussion

Antibodies can potentially target any epitope of a given antigen, thanks to the extremely high variability in the repertoire of B-cell clones. Despite this potential, epitope specificities are not equally targeted by humoral responses, with the most frequently targeted epitopes defined as immunodominant. Viruses have evolved different strategies to escape Ab neutralization, among others (hyper)mutation of viral antigens. Antibodies able to neutralize multiple viral variants are defined as broadly neutralizing, and they represent the ultimate target of most vaccination strategies. BnAbs represent a fundamental tool to mount effective protection against highly mutating viruses, such as influenza and HIV. Despite the prolonged efforts to induce such humoral responses upon vaccination, bnAbs are generally highly subdominant when compared to other strain-specific antibodies.

Understanding the rules defining immunodominance is of paramount importance to improve the design of future vaccination strategies. In recent decades, several hypotheses have been suggested to explain the molecular mechanisms underlying the inconsistent induction of bnAbs, including antigen mutation, epitope accessibility, high BCR mutational load required, B-cell precursor frequencies, and HLA class II polymorphisms. Nevertheless, none of these hypotheses fully recapitulates the counter-selection of bnAbs, especially in the context of isolated antigens in vaccination studies. Pioneering work by Berzofsky and Celada demonstrated that BCR or antibody binding to antigens could boost or inhibit the presentation of specific CD4 epitopes based on their relative positioning. Indeed, antibodies can both sterically hinder and inhibit the presentation of CD4 T cell epitopes located within the Ab-bound region or stabilize adjacent epitopes to facilitate their mounting on HLA molecules, thus enhancing their presentation.

In the present work, we propose the theory of immunodominance relativity, according to which the relative positioning of B and T cell epitopes within an antigen shapes immunodominance. Indeed, we found that subdominant conserved regions targeted by bnAbs (e.g., CD4 binding site of HIV gp120 or the stem of influenza HA) often overlap with clusters of highly immunodominant CD4 epitopes. In support of this observation, anti-CD4 binding site and anti-HA stem Abs inhibit antigen presentation upon binding, reinforcing the idea that overlapping B and T cell epitopes can lead to the inhibition of antigen presentation. Our analysis suggests that bnAb B cell precursors specific for conserved regions of HIV and influenza inhibit the presentation of immunodominant CD4 determinants upon BCR binding, resulting in poor T cell help during the GC reaction and consequent counter-selection in favor of other dominant B cell clones. On the contrary, non-neutralizing or strain-specific immunodominant B cell precursors may boost the presentation of adjacent immunodominant CD4 epitopes, resulting in increased T cell help and selective advantage during GC reactions. It is worth highlighting that the mechanism we proposed in this study might have different impacts on B cell immunodominance depending on the B epitopes analyzed. Indeed, specific bnAb-targeted B cell determinants may be subdominant as a function of other immunosuppressive mechanisms, which are extensively discussed above. Interestingly, the potent anti-HIV bnAb antibody can eventually be isolated from infected patients years after the infection. A possible explanation for this event in light of our proposed theory is that extensive T–B crosstalk and BCR mutational load in GCs over the years might overcome defects in antigen presentation by bnAb-bearing B cell clones, as the impairment in antigen presentation is significant but not absolute. We would also like to point out that a conformational (rather than linear) epitope analysis might impact epitope interactions and will be of great interest to obtain in this context. Nevertheless, our analysis likely comprises both linear and conformational epitopes, as the immunodominant B regions provided likely contained both linear and structural determinants.

This working hypothesis currently lacks formal experimental demonstration; however, we built the model gathering data from a multitude of independent publications over the last four decades and found supportive experimental evidence published by different groups. Finally, we propose that the relative positioning of B–T epitopes may be one additional mechanism that cooperates with the other above-mentioned processes to influence immunodominance. If demonstrated, this theory can improve the understanding of the immune responses against current and future pandemics and will indicate a rational way to design antigens for effective vaccination strategies.

## 5. Materials and Methods

### 5.1. CD4 and B-Cell Immunodominant Epitope Selection and Analysis

To focus the analysis on HLA-independent immunodominant determinants, experimentally validated CD4 and B epitopes that were found to be dominant in at least two different species/mouse strains or multiple patients were selected for further testing. B-cell epitopes were arbitrarily classified as distant from (>15aa), adjacent to (<15aa), or overlapping (<0aa) with dominant CD4 epitopes based on their distance and positioning within the antigenic linear sequence. As such, peptides distant less than 15aa were considered as likely to be near enough for the Ab to suppress the CD4 epitope.

### 5.2. Crystal Structures’ Visualization

To highlight the inter-positioning of B- and CD4-dominant epitopes in 3D structures, the above-mentioned PDB files were modified using Pymol (2.3.5). CD4-dominant epitopes were labeled in blue, B-dominant epitopes in red, and overlapping regions in magenta. Amino acids essential for bnAb binding to antigens are listed in the indicated figures, and those overlapping with CD4-dominant epitopes are highlighted in magenta.

### 5.3. Statistical Analysis

Prism software (GraphPad 9.0.1) was used for all statistical analysis. A chi-squared test was performed to evaluate the relationship between immunogenicity and the relative distribution of B- and T-cell epitopes within antigens. *p* < 0.05 was considered significant. In the summary graphs, the points indicate samples and horizontal lines are the means. Error bars indicated standard error mean (SEM).

## Figures and Tables

**Figure 1 vaccines-10-01227-f001:**
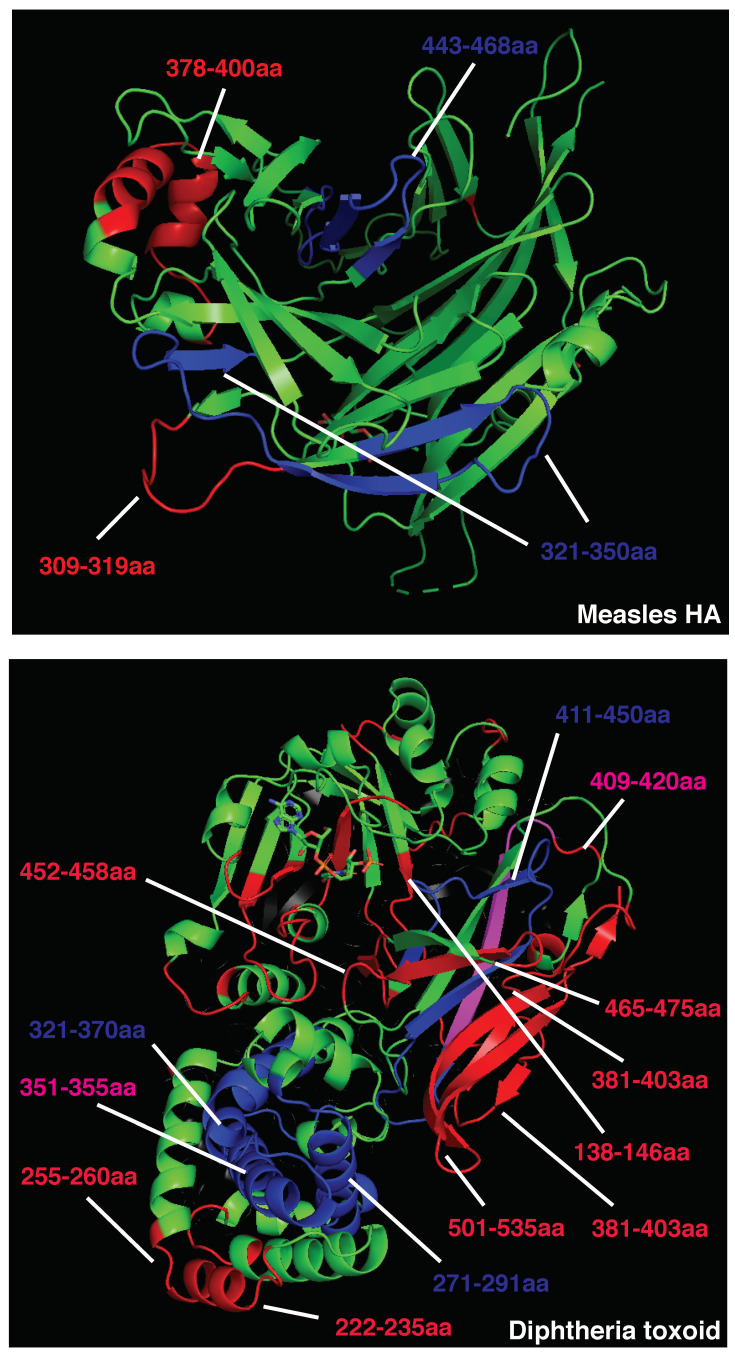
Crystal structure of measles hemagglutinin (PDB:2ZB6, top) or diphtheria toxoid (PDB:1MDT, bottom). Immunodominant B cell epitopes are depicted in red, while dominant CD4 determinants are highlighted in blue. Overlapping epitopes are highlighted in pink.

**Figure 2 vaccines-10-01227-f002:**
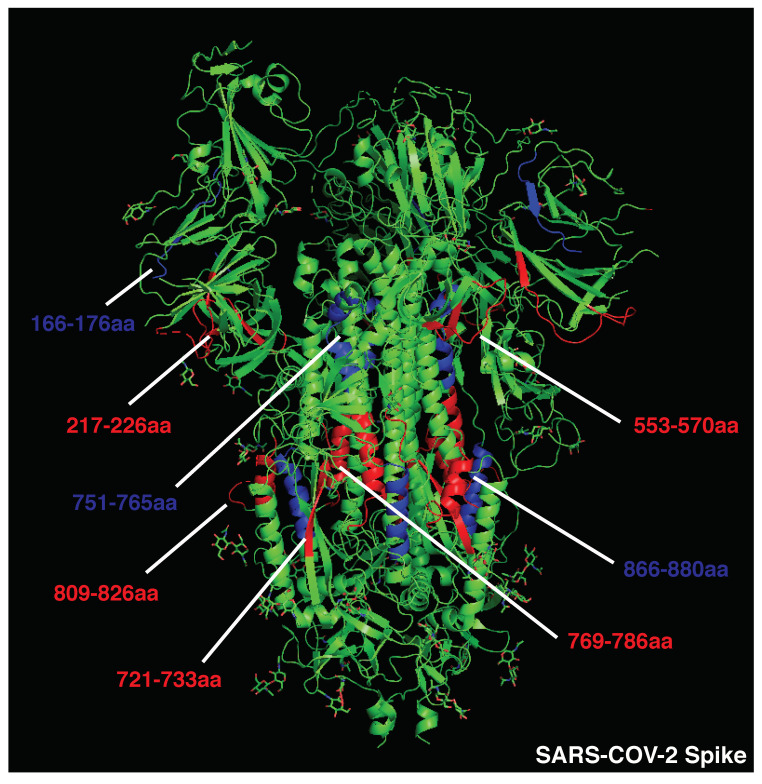
Crystal structure of SARS-CoV-2 spike protein (PDB:6VSB). Immunodominant B cell epitopes are depicted in red, while dominant CD4 determinants are highlighted in blue.

**Figure 3 vaccines-10-01227-f003:**
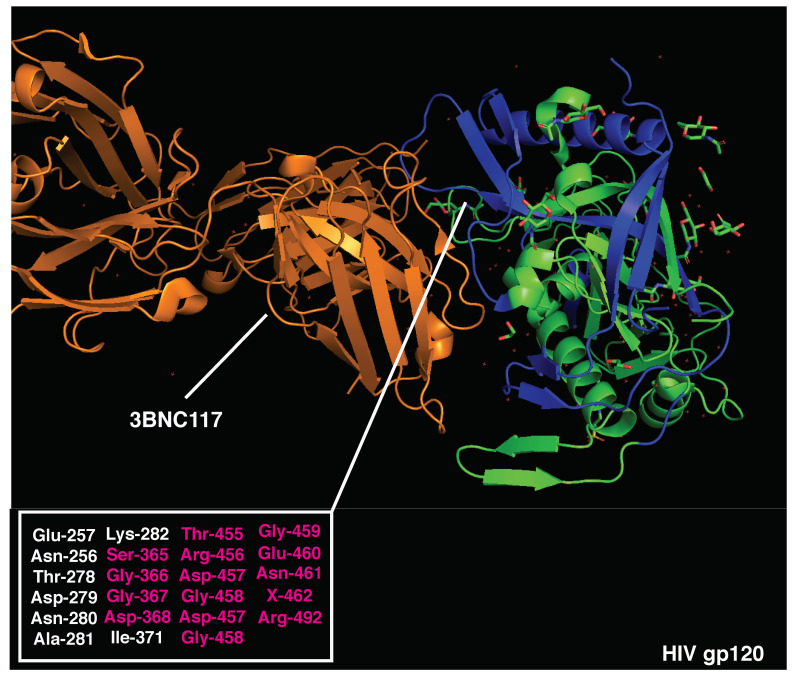
Crystal structure of HIV gp120 monomer and 3BNC117 bnAb (PDB:4JPV). Immunodominant CD4 epitopes are highlighted in blue. Amino acids essential for 3BNC117 binding are listed (bottom left) and those overlapping with CD4-dominant regions are highlighted in red.

**Figure 4 vaccines-10-01227-f004:**
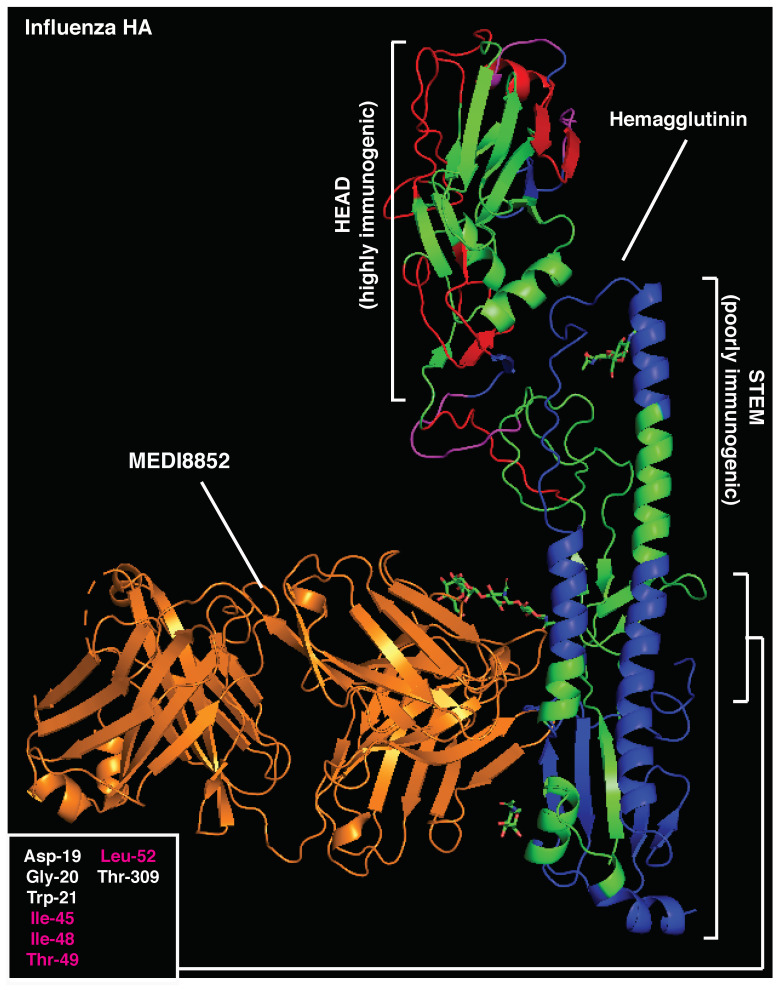
Crystal structure of influenza hemagglutinin and MEDI8852 bnAb (PDB:5JW3). Immunodominant B-cell epitopes are depicted in red, while dominant CD4 determinants are highlighted in blue. Amino acids essential for MEDI8852 binding are listed (bottom left) and those overlapping with CD4-dominant regions are highlighted in red.

**Figure 5 vaccines-10-01227-f005:**
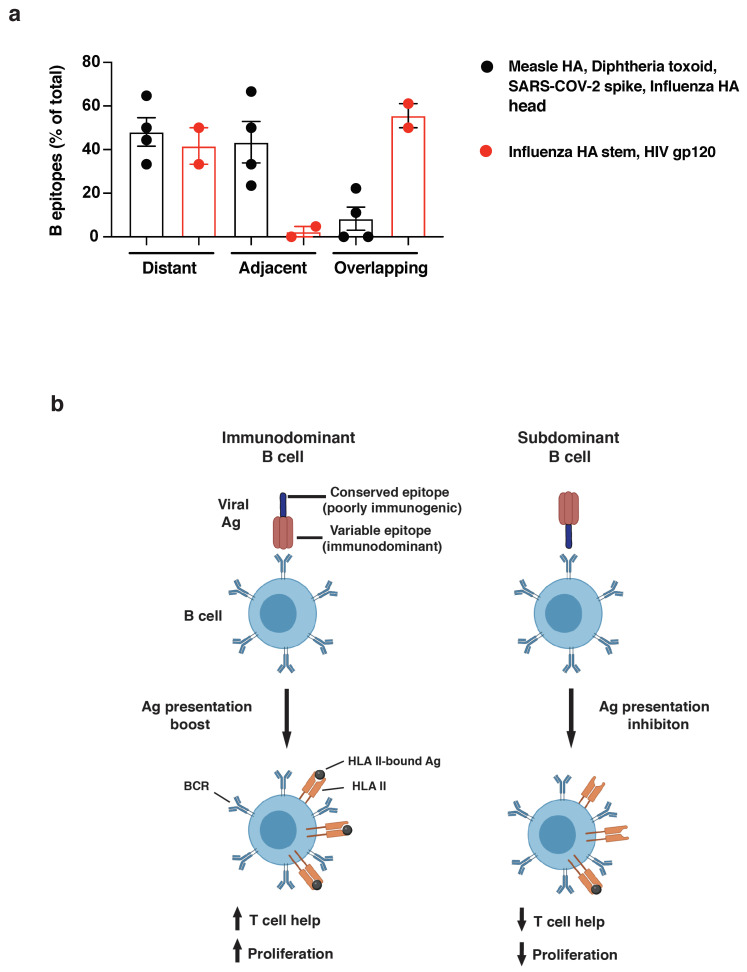
(**a**) Quantification of dominant B cell epitopes classified as distant from (>15aa), adjacent to (<15aa), or overlapping with (<0aa) immunodominant CD4 determinants within the same antigen. Good nAb inducers (measles HA, diphtheria toxoid, SARS-CoV-2 spike, and influenza HA head: black dots) and poor nAb inducers (influenza HA stem and HIV gp120: red dots) are compared. Chi-squared analysis was applied (see Appendix A). (**b**) Model figure showing immunodominant B cell (left) recognizing immunogenic viral regions and boosting antigen presentation upon BCR binding; right, subdominant B-cell clone binding conserved/poorly immunogenic viral regions and inhibiting presentation of overlapping CD4 epitopes.

## Data Availability

All data are available in the manuscript text, figures or tables.

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
