# Peer review of "The Relative Positioning of B and T Cell Epitopes Drives Immunodominance"

_vaccines, 2022, doi:10.3390/vaccines10081227_

Round 1

Reviewer 1 Report

In this perspective piece, the authors make a case for a major role of epitope positioning within an antigen with respect to the likelihood of developing broadly neutralizing antibodies (bnAb). Based on a study of the location of peptides that evoke MHC CII-dependent CD4 T cell responses and the regions of the same protein that evoke antibody responses, especially bnAb to viruses like HIV and influenza, they report that sequence overlap between conserved regions of viral envelope/capsid proteins that are major bnAb binding sites and the location of major epitopes for CD4 T cell responses is anti-correlated with generation of bnAb responses in infected or vaccinated individuals. Given this anti-correlation, they suggest that the difficulty in producing bnAb is more linked to this feature of epitope overlap than to other explanations of immunodominance such as physical inaccessibility of a site to BCR binding or the number of mutations needed to generate the bnAb from the germline BCR sequence.

While the influence of antibody binding to an antigen on the processing of that antigen and the subsequent availability of pMHC ligands for CD4 T cells is well documented in the literature (as the authors note), I am not aware of a focus on antibody footprint and CD4 epitope location in limiting the generation of bnAb to important viruses. For this reason, the manuscript is of value to the field. The examples provided by the authors are consistent with their model, although as they note, without experiments in which one changes the positioning of the same epitopes within a protein and analyze the ensuing responses under controlled conditions, the concept remains theoretical rather than demonstrated.

There are a few improvements that I can suggest to the submission.

1.     It is not clear why viruses should evolve to have potent CD4 T cell epitopes in the regions that could lead to bnAb. It would seem beneficial to the virus to evolve away from having such epitopes at all. Of course, MHC polymorphism and the multiple loci and alleles in a single individual make such sequence evolution difficult, especially in conserved regions where sequence diversity is limited by potential loss of viral fitness, but the authors don’t discuss this issue adequately in my view. It would be valuable to consider whether the location of CD4 epitopes in such regions is a result of the limitation of sequence diversity or it there is actually a benefit to the virus to position good T cell epitopes in that location. One possibility is that BCR binding elsewhere on the protein would favor induction of T cells to these epitopes and enhance help for the irrelevant B cell target sites, which in turn would further limit bnAb responses to the relevant site through a competitive mechanism. This is hinted at in the text but should be explicitly discussed.

2.     Crotty and Sette have done extensive studies of the specificity of responses to SARS-Cov2 that are not cited here – see

                     Cell. 2022 Mar 3;185(5):847-859

                   Cell Host Microbe. 2021 Jul 14;29(7):1076-1092

                   Cell. 2021 Feb 18;184(4):861-880

                   Cell. 2020 Nov 12;183(4):996-1012

among others. These take into account MHC polymorphism, something that is a concern for the main data in this paper. The epitopes illustrated have only been docuemented for a limited number of HLA alleles and since the virus needs to evolve to handle the diverse alleles in a large population, it would be surprising that all epitopes acting as effective MHC CII-presented ligands for such a diversity of alleles would wind up at the bnAb-target site. Said another way, how many HLA allotypes have been examined in reaching these definitions of CD4 epitopes? It is hard to imagine that the same peptide(s) are well presented by diverse CII alleles, although some such alleles would be present in many individuals, at least in some human populations, and hence a general 'rule' could be imagined. The authors should discuss this effect of polymorphisms more extensively and provide specific information on whether the mapped epitopes in the figures are presented by highly prevalent alleles in racially different populations.

A few minor points:

1.     lines 57-58: HIV but NOT influenza hypermutates. Most influenza antigenic changes are due to reassortment and gradual antigenic drift.

2.     Line 20 – immunogenic refers to activating the adaptive immune system, not to whether a particular epitope is immunodominant – remove this term here.

3.     Line 192 – SARS-Cov2 does NOT produce long lasting antibody responses

Author Response

Reviewer 1

In this perspective piece, the authors make a case for a major role of epitope positioning within an antigen with respect to the likelihood of developing broadly neutralizing antibodies (bnAb). Based on a study of the location of peptides that evoke MHC CII-dependent CD4 T cell responses and the regions of the same protein that evoke antibody responses, especially bnAb to viruses like HIV and influenza, they report that sequence overlap between conserved regions of viral envelope/capsid proteins that are major bnAb binding sites and the location of major epitopes for CD4 T cell responses is anti-correlated with generation of bnAb responses in infected or vaccinated individuals. Given this anti-correlation, they suggest that the difficulty in producing bnAb is more linked to this feature of epitope overlap than to other explanations of immunodominance such as physical inaccessibility of a site to BCR binding or the number of mutations needed to generate the bnAb from the germline BCR sequence.

While the influence of antibody binding to an antigen on the processing of that antigen and the subsequent availability of pMHC ligands for CD4 T cells is well documented in the literature (as the authors note), I am not aware of a focus on antibody footprint and CD4 epitope location in limiting the generation of bnAb to important viruses. For this reason, the manuscript is of value to the field. The examples provided by the authors are consistent with their model, although as they note, without experiments in which one changes the positioning of the same epitopes within a protein and analyze the ensuing responses under controlled conditions, the concept remains theoretical rather than demonstrated.

We thank the Reviewer for appreciating our effort to emphasize the possible relationship between epitope positioning and immunodominance in the context of bnAb production. While we strongly agree with the Reviewer that formal experiments will help supporting our theory, this effort is beyond the scope of this Hypothesis paper and will be the focus of our future studies.

There are a few improvements that I can suggest to the submission.

  1. It is not clear why viruses should evolve to have potent CD4 T cell epitopes in the regions that could lead to bnAb. It would seem beneficial to the virus to evolve away from having such epitopes at all. Of course, MHC polymorphism and the multiple loci and alleles in a single individual make such sequence evolution difficult, especially in conserved regions where sequence diversity is limited by potential loss of viral fitness, but the authors don’t discuss this issue adequately in my view. It would be valuable to consider whether the location of CD4 epitopes in such regions is a result of the limitation of sequence diversity or it there is actually a benefit to the virus to position good T cell epitopes in that location. One possibility is that BCR binding elsewhere on the protein would favor induction of T cells to these epitopes and enhance help for the irrelevant B cell target sites, which in turn would further limit bnAb responses to the relevant site through a competitive mechanism. This is hinted at in the text but should be explicitly discussed.

We thank the Reviewer for this insightful comment. We think there are several reasons about why viruses able to induce strong and early nAbs may have evolved to display potent CD4 T cell epitopes that favor this process. As the Reviewer suggests, this may be linked to limited sequence diversity within important viral regions, which are not accessible to mutations because of loss in viral fitness. This possibility, however, would not explain why such conserved regions should contain dominant CD4 epitopes that supports nAb production. One additional possibility proposed by the Reviewer, is that these epitopes may be boosted for Ag presentation by irrelevant BCRs, thus competing with nAb-bearing B cell clones. This competitive mechanism, however, is not enough to block the early nAb production against such viruses, but may be needed to diversify humoral responses (as discussed in the manuscript). We believe that certain viruses may have passively evolved to maintain such immunodominant regions to obtain long-term fitness advantages. This might be particularly important in the case of cytopathic viruses, including VSV and Measles. These cytopathic viruses are able to induce widespread cell damage and can generally infect a new host prior to the generation of antiviral adaptive immune responses, while non-cytopathic viruses such as HIV must escape long-term humoral immunity. Early and potent neutralization of cytopathic viruses is key for reaching an evolutionary balance in host-pathogen interaction, given that hosts incapable of them are likely to succumb to infection. Accordingly, if a cytopathic virus does not induce early and potent nAb responses in the host population, this host-pathogen interaction is more likely to disappear in the long-term. Thus, we believe that aggressive and fast replicating cytopathic viruses have evolved to induce protective Ab response in the infected hosts to reach host-pathogen interaction balance and increase their fitness. To better explain these concepts in the main text, we have added the following (highlighted in yellow in the revised text): “The presence of conserved and dominant viral CD4 epitopes that support nAb production raises the intriguing question about why such viruses have evolved to maintain these immune determinants. Some of these epitopes may be part of protein regions that do not allow for high mutational load, as it would result in loss of viral fitness. However, this possibility would not explain why such conserved regions contain dominant CD4 epitopes. A second possibility is that these viruses maintained these conserved and dominant CD4 epitopes to obtain long-term fitness advantages. Indeed, this may be particularly the case of aggressive and fast replicating cytopathic viruses (including VSV and Measles), which are highly infectious and induce widespread cell damage. Early and potent neutralization of such viruses is key for reaching a balance in host-pathogen interaction, given that hosts incapable of neutralizing such viruses are likely to succumb to infection. In addition, these viruses usually escape Ab responses by infecting a new host prior to the generation of antiviral adaptive immunity and thus are not highly affected by neutralization. For these reasons, these cytopathic viruses were not likely selected to evolve a protective mechanism against the pressure of the adaptive immune system.

  1. Crotty and Sette have done extensive studies of the specificity of responses to SARS-Cov2 that are not cited here – see

Cell. 2022 Mar 3;185(5):847-859

Cell Host Microbe. 2021 Jul 14;29(7):1076-1092

Cell. 2021 Feb 18;184(4):861-880

Cell. 2020 Nov 12;183(4):996-1012

among others. These take into account MHC polymorphism, something that is a concern for the main data in this paper. The epitopes illustrated have only been docuemented for a limited number of HLA alleles and since the virus needs to evolve to handle the diverse alleles in a large population, it would be surprising that all epitopes acting as effective MHC CII-presented ligands for such a diversity of alleles would wind up at the bnAb-target site. Said another way, how many HLA allotypes have been examined in reaching these definitions of CD4 epitopes? It is hard to imagine that the same peptide(s) are well presented by diverse CII alleles, although some such alleles would be present in many individuals, at least in some human populations, and hence a general 'rule' could be imagined. The authors should discuss this effect of polymorphisms more extensively and provide specific information on whether the mapped epitopes in the figures are presented by highly prevalent alleles in racially different populations.

We thank the Reviewer for this important comment. We have included the suggested references as indicated, as we think these are important manuscripts to be cited. As mentioned in the main text and supplementary tables, to restrict our analysis to ‘dominant’ CD4 epitopes - that are more likely to be HLA-independent - we have selected determinants that have been validated as dominant in either different animal strains, animal species, or several patients. For example:

-SARS-COV-2 immunodominant CD4 epitopes were validated in several patients (n=42) (Yanchun Peng et al., Nat Imm 2020), and epitopes presented by 6-10/42 patients were selected as immunodominant;

-for HIV-gp120, the selected CD4 epitopes were chosen among determinants with the following features:

-shared by at least 2 different mouse strains (CBA/J, BALBc, BL/6 mice; Guixiang Dai,N. et al., JBC 2001; Surman S et al., PNAS 2001)

- shared by at least 2 different species (mice, macaques, humans; Surojit Sarkar et al., JI 2002; Jay A. Berzofsky et al., Nature 1988; R D schrier et al., JI 1989; B Wahren et al., J Acquir Immune Defic Syndr 1988)

- shared by multiple analyzed macaques (n=8) or patients (n=14-40).

-for Influenza-HA, dominant CD4 epitopes were obtained by determinants found in immunized patients (n=4-30, Antonino Cassotta et al., JEM 2020; Samuel J. Landry., Journal of Virology 2008) and two mouse strains (Samuel J. Landry., Journal of Virology 2008).

Thus, given that our analysis was restricted to experimentally validated CD4 epitopes shared by at least two different mouse strains, animal species, and/or multiple individuals, we believe that our findings are unlikely to be strongly affected by HLA allele-dependent effects. In addition, we respectfully disagree with the Reviewer on the following point “It is hard to imagine that the same peptide(s) are well presented by diverse CII alleles”. Indeed, while single epitopes may be more or less efficiently loaded on different HLAII alleles, clusters of immunodominant CD4 epitopes have been described for the presented antigens, particularly those that escape broad neutralization (as mentioned in the main text). In addition, some of the viral epitopes that we have selected have been validated to be immunodominant in multiple macaques or human patients (ranging from 10-100% of the analyzed cohorts). This suggests that most of the selected epitopes are unlikely to be immunodominant for a restricted set of HLAII alleles.

A few minor points:

  1. lines 57-58: HIV but NOT influenza hypermutates. Most influenza antigenic changes are due to reassortment and gradual antigenic drift.

We thank the Reviewer for this comment. We have corrected the main text accordingly.

  1. Line 20 – immunogenic refers to activating the adaptive immune system, not to whether a particular epitope is immunodominant – remove this term here.

We thank the Reviewer for this comment. Here we are using the term immunogenic to refer to protein regions able to evoke humoral responses (not immunodominant).

  1. Line 192 – SARS-Cov2 does NOT produce long lasting antibody responses

We thank the Reviewer for this comment. We have corrected the main text accordingly.

Reviewer 2 Report

Based on the current literature and the analysis of crystal structures, the authors hypothesise that the relative positioning of B and T cell epitopes could be one additional mechanism to influence immunodominance. This hypothesis can be potentially exciting, but the reviewer would encourage the authors to supplement some key wet-lab data that could support the hypothesis.

Author Response

Reviewer 2

“Based on the current literature and the analysis of crystal structures, the authors hypothesise that the relative positioning of B and T cell epitopes could be one additional mechanism to influence immunodominance. This hypothesis can be potentially exciting, but the reviewer would encourage the authors to supplement some key wet-lab data that could support the hypothesis”.

We thank the Reviewer for this comment. While we agree with the Reviewer that wet-lab data will be key to support and demonstrate our hypothesis, we think this is beyond the scope of this Theory manuscript and will be the focus of our future experimental effort. To compensate for the lack of novel experimental evidence, we have cited key experiments performed by other groups that show relevant impairment of antigen presentation in HIV and influenza when bound by nAbs. In addition, we believe that these experiments provide a stronger support for our hypotheses than proof-of-concept wet-lab data generated by us, since these groups are totally independent from us and performed their experiments to demonstrate other hypotheses.

Reviewer 3 Report

This study proposed a hypothesis that the relative positioning of B and T cell epitopes drives immunodominance. I should appreciate the authors' time and patience to come up with some results. However, there are still several problems that deduct from the quality of this manuscript. Below are several comments on this work. 

1. Could you add more significance test methods to your results?

2. How did you verify your hypothesis in wet lab?

Author Response

Reviewer 3

“This study proposed a hypothesis that the relative positioning of B and T cell epitopes drives immunodominance. I should appreciate the authors' time and patience to come up with some results. However, there are still several problems that deduct from the quality of this manuscript. Below are several comments on this work. 

  1. Could you add more significance test methods to your results?

We thank the Reviewer for these comments. We believe the Chi-square analysis to be the most adequate statistical test to analyze the contingency of relative positioning of B cell and CD4 T cell epitopes and Ab immunodominance. Other more commonly used statistical tests, including multiple non-parametric t test and Kruskal-Wallis (we cannot assume Gaussian distribution of epitopes since they are clustered in HIV and influenza), would also provide us with statistically significant differences, but may not be the best choice since they would only evaluate the difference in the distribution of epitopes rather than its association with immunodominance.

  1. How did you verify your hypothesis in wet lab?”

We are planning to design experiments aiming to experimentally test our hypothesis, including but not limited to the design of transgenic immunogens with dominant CD4 epitopes differently positioned within their sequences.

Round 2

Reviewer 2 Report

Understood that this is a Hypothesis manusciprt: in such case, the authors are advised to modify the tone of the manuscript throughout, i.e. using a less definitive tone. Plus, the authors should include concrete plans to experimentally verify the hypothesis.

Author Response

Please see the attached PDF (complete Rebuttal).

-We thank the Reviewer for the comments. We strongly highlight that this is an undemonstrated theory that requires experimental validation throughout the text (some examples listed in the attachment). Accordingly, we respectfully disagree with the Reviewer about the need to further modification of the tone of the manuscript throughout.

-We are in the process of designing a detailed experimental approach to verify our hypothesis. Indeed, we have applied for grant funding to support the generation of knock-in BCR transgenic murine lines that we think are crucial to answer our questions. To study the influence of BCR specificities on GC reactions, we will compare in vivo activation, selection and differentiation of transferred B cells that bear either gp120/HA-specific bnAbs or non-neutralizing receptors, within the same immunized host. We will also sort bnAb and non-neutralizing receptor-bearing B cells at different time points after immunization, to study the MHC2-related peptidome in the presence of different receptor specificities, by mass spectrometry. To test whether epitope positioning has an impact on Ag-presentation, we will generate recombinant viral proteins (e.g. gp120, HA) in which the same dominant CD4 epitope(s) are differently positioned across the linear protein sequence. These recombinant proteins will be than used to study Ag-presentation in vitro and used to immunized control and BCR tg mice in vivo. Nevertheless, we think that sharing a detailed experimental proposal in the main text is beyond the scope of this theory paper. The validity of a theory, indeed, does not depend on whether experimental tools and solutions are readly available to test it. In addition, we would like to keep our experimental approach(es) as confidential as possible, given we have submitted grant proposals about these concepts and theory. We are providing here the approaches that we have now in mind to convince the Reviewer that our hypothesis can be experimentally validated in the future, but we prefer not to share these details on the main text.

Please see the attached PDF.
